# Anti-Inflammatory Activity of Fucoidan Extracts In Vitro

**DOI:** 10.3390/md19120702

**Published:** 2021-12-11

**Authors:** Tauseef Ahmad, Mathew Suji Eapen, Muhammad Ishaq, Ah Young Park, Samuel S. Karpiniec, Damien N. Stringer, Sukhwinder Singh Sohal, J. Helen Fitton, Nuri Guven, Vanni Caruso, Rajaraman Eri

**Affiliations:** 1School of Health Sciences, University of Tasmania, Launceston, TAS 7248, Australia; tauseef.ahmad@utas.edu.au; 2Respiratory Translational Research Group, Department of Laboratory Medicine, School of Health Sciences, College of Health and Medicine, University of Tasmania, Launceston, TAS 7248, Australia; mathew.eapen@utas.edu.au (M.S.E.); sukhwinder.sohal@utas.edu.au (S.S.S.); 3School of Pharmacy and Pharmacology, University of Tasmania, Hobart, TAS 7005, Australia; Muhammad.Ishaq@utas.edu.au (M.I.); nuri.guven@utas.edu.au (N.G.); vanni.caruso@utas.edu.au (V.C.); 4Marinova Pty Ltd., Cambridge, TAS 7170, Australia; ahyoung.park@marinova.com.au (A.Y.P.); sam.karpiniec@marinova.com.au (S.S.K.); damien.stringer@marinova.com.au (D.N.S.); 5RDadvisor, Hobart, TAS 7006, Australia; 6ISAL Foundation, Research on Pain, Torre Pedrera, 204-47922 Rimini, Italy

**Keywords:** fucoidan, PBMCs, THP-1, TNF-α, IL-1β, and IL-6, *Undaria pinnatifida*, *Fucus vesiculosus*, *Macrocystis pyrifera*, *Ascophyllum nodosum*, *Laminaria japonica*

## Abstract

Fucoidans are sulfated, complex, fucose-rich polymers found in brown seaweeds. Fucoidans have been shown to have multiple bioactivities, including anti-inflammatory effects, and are known to inhibit inflammatory processes via a number of pathways such as selectin blockade and enzyme inhibition, and have demonstrated inhibition of inflammatory pathologies *in vivo*. In this current investigation, fucoidan extracts from *Undaria pinnatifida, Fucus vesiculosus, Macrocystis pyrifera, Ascophyllum nodosum,* and *Laminaria japonica* were assessed for modulation of pro-inflammatory cytokine production (TNF-α, IL-1β, and IL-6) by human peripheral blood mononuclear cells (PBMCs) and in a human macrophage line (THP-1). Fucoidan extracts exhibited no signs of cytotoxicity in THP-1 cells after incubation of 48 h. Additionally, all fucoidan extracts reduced cytokine production in LPS stimulated PBMCs and human THP-1 cells in a dose-dependent fashion. Notably, the 5–30 kDa subfraction from *Macrocystis pyrifera* was a highly effective inhibitor at lower concentrations. Fucoidan extracts from all species had significant anti-inflammatory effects, but the lowest molecular weight subfractions had maximal effects at low concentrations. These observations on various fucoidan extracts offer insight into strategies that improve their efficacy against inflammation-related pathology. Further studies should be conducted to elucidate the mechanism of action of these extracts.

## 1. Introduction

Fucoidans are a class of sulfated, fucose-rich polymers present in the matrix of various species of brown seaweed, including *Cladosiphon* sp. (mozuku), *Laminaria japonica* (kombu), *Fucus vesiculosus* (bladderwrack), and *Undaria pinnatifida* (wakame) [1,2]. Recent research *in vitro* and *in vivo* revealed numerous biological activities of fucoidan [3,4,5,6,7]. The reported biological activities of fucoidan are diverse in nature and include anti-tumour and immunomodulation [8,9], anti-coagulant [10], inhibition of enzymes [11], blocking of lymphocyte adhesion and invasion [12,13], and anti-inflammatory effects [14]. These distinctive properties make fucoidan a potential nutraceutical and pharmaceutical candidate in the fields of disease prevention and drug development [14,15,16,17]. However, the characteristics and efficacy of these biological activities depend on the composition, source, molecular weight, purity level, and structure of fucoidan and can vary from species to species [3].

Anti-inflammatory and immunomodulation properties are key features of fucoidan. The potential targets for fucoidan are immune cells, including macrophages, natural killer cells, dendritic cells, and lymphocytes [3,18,19,20]. Several previous reports have demonstrated the anti-inflammatory activities of fucoidan and fucoidan-containing extracts in different experimental models *in vitro* and *in vivo* [2,3,21,22]. One of the previous *in vivo* studies from our research group revealed that dietary fucoidan extracts from *Undaria pinnatifida* and *Fucus vesiculosus* were highly effective in ameliorating dextran sulfate sodium (DSS)-induced colitis in mice through a consistent downregulation of a significant number of pro-inflammatory cytokines, including TNF-α, IL-1β, and IL-6 [14]. It is now generally recognized that the inflammatory response in patients and animal models of inflammatory bowel disease (IBD) is primarily macrophage driven [14,21,22,23]. In the event of intestinal inflammation, monocytes are recruited and differentiated into macrophages within the lamina propria [14,22]. It is considered that the initial exposure of interstitial macrophages to bacterial antigens is responsible for the activation of macrophages and that, at least in the DSS model, these activated macrophages subsequently stimulate the proliferation of T cells [14]. Macrophages play a crucial role in stimulating and modulating the immune response (14). Furthermore, they regulate the function of adaptive immunity through cell-to-cell interaction or fluid-phase modulation via chemokines and cytokines [24]. The biological activities of macrophages, including phagocytosis, migration, and cytokine secretion, are critical to the immune response outcome [25].

Following on from our previous *in vivo* findings on a colitis model using fucoidan from *Undaria pinnatifida* and a fucoidan-polyphenol extract from *Fucus vesiculosus* [14], this present study investigates the effect of multiple fucoidan extracts on lipopolysaccharide (LPS) induced whole human PBMC and THP-1 monocyte cell lines on pro-inflammatory cytokine production. Fucoidan from five different sources has been used in the current investigation, including *Undaria pinnatifida* (UPF), *Fucus vesiculosus* (FVF), *Macrocystis pyrifera* (MPF), *Ascophyllum nodosum* (ANF), and *Laminaria japonica* (LJF). Extracts from UPF, FVF, and MPF were further depyrogenated and/or fractionated to yield depyrogenated (DP) and low molecular weight fucoidan extracts (LMWF). The source, purity, and molecular weight for each extract are described in Table 1. Here, we aimed to compare the anti-inflammatory effects of these extracts with a primary focus on major pro-inflammatory markers, including TNF-α, IL-1β, and IL-6 and assess the dose-dependent inhibition of these cytokines after pre-treatment with fucoidan extracts. Additionally, we also sought to investigate the cytotoxic effects of these fucoidan extracts in our experimental model.

## 2. Results

### 2.1. Effect of Fucoidan Extracts on Cell Viability

The MTT cell proliferation assay is commonly used to measure cytotoxicity or cell proliferation [26]. It is centered on the metabolic production of NAD(P)H, which is used as a surrogate marker for cell death, cytostatic activity, or metabolic inhibition by test compounds [26,27]. To assess the short-term effects of fucoidan treatment, THP-1 cells were exposed to fucoidan extracts at various concentrations (0, 10, 50, 100, and 200 µg/mL) for a period of 48 h (Figure 1). Overall, fucoidan extracts exhibited no signs of cytotoxicity in THP-1 cells after incubation of 48 h.

### 2.2. Effects of UPF and FVF in LPS Induced TNF-α, IL-1β, and IL-6 in Human PBMCs

To evaluate the inhibitory effects of UPF and FVF on pro-inflammatory cytokines (TNF-α, IL-1β, and IL-6) in LPS-stimulated human PBMCs, PBMCs were first pre-treated with UPF and FVF at three different concentrations (10, 50, and 100 µg/mL) for 1 h followed by the addition of (0.5 mg/mL) LPS and incubated for 24 h. The cell culture supernatants were evaluated through sandwich ELISA to measure TNF-α, IL-1β, and IL-6 levels. As expected, LPS stimulation significantly elevated TNFα, IL-1β, and IL-6 secretion in the cultured human PBMCs in the absence of fucoidan. UPF significantly reduced the induction of TNF-α and IL-6 by LPS stimulation in a dose-dependent manner (Figure 2A,C); however, the levels of IL-1β were not significantly altered by the pre-treatment of UPF (Figure 2B). In contrast to UPF, FVF pre-treatment significantly inhibited the levels of all three pro-inflammatory cytokines, TNF-α, IL-1β, and IL-6, in LPS stimulated human PBMCs (Figure 2D–F). These results suggest that fucoidan from *Undaria pinnatifida* and *Fucus vesiculosus* effectively suppresses pro-inflammatory cytokine production by altering the production levels of TNF-α, IL-1β, and IL-6 in human PBMCs.

### 2.3. Effect of Fucoidan Extracts on LPS Induced TNF-α Pro-Inflammatory Cytokine in Human THP-1 Monocytes

#### 2.3.1. *Undaria pinnatifida* Fucoidan (UPF) and Its Derived Extracts Suppress LPS Induced TNF-α in Human THP-1 Cells

Three extracts of *Undaria pinnatifida* (UPF, DP UPF, and 5–30 DP UPF) were used in this study to evaluate their anti-inflammatory effects. These extracts differ based on the difference in their molecular weights, as indicated in Table 1. UPF is a whole, nutritional grade extract with broad molecular weight distribution. The extract DP UPF has been depyrogenated, while 5–30 DP UPF has been further processed to produce a narrower molecular weight, ranging from 5–30 kDa. Pre-treatment of human THP-1 cells with UPF, DP UPF, and 5–30 DP UPF followed by induction with 0.5 µg/mL of LPS for 24 h demonstrated inhibition of LPS induced TNF-α in a dose-dependent manner. Nutritional grade UPF has shown dose-dependent inhibition with 70% (*p* < 0.0001), 50% (*p* < 0.0001), 40% (*p* < 0.001), and 30% (*p* < 0.05) inhibition at 200 µg/mL, 100 µg/mL, 50 µg/mL, and 10 µg/mL respectively (Figure 3A). In contrast, 5–30 DP UPF inhibited the LPS induced TNF-α in THP-1 monocytes effectively at lower concentrations with a percentage inhibition of 20% (*p* < 0.001), 15% (*p* < 0.001), and 10% (*p* < 0.05) at concentrations of 10 µg/mL, 50 µg/mL, and 100 µg/mL, respectively. (Figure 3C). However, DP UPF only showed a trend of inhibition of LPS induced TNF-α with 10% inhibition (*p* < 0.05) at 100 µg/mL concentration (Figure 3B). These data suggest that the nutritional grade of *Undaria pinnatifida* fucoidan (UPF) possesses strong anti-inflammatory properties across all concentrations used in the current study; however, the range of 5–30 DP UPF shows strong inhibition at lower concentrations compared to DP UPF.

#### 2.3.2. *Fucus vesiculosus* Fucoidan (FVF) and Its Derived Extracts Suppress LPS Stimulated TNF-α in Human THP-1 Cells

*Fucus vesiculosus* (FVF) has shown a similar trend to that of nutritional grade UPF and exhibited strong dose dependant inhibition of LPS induced TNF-α in THP-1 monocytes with 60% (*p* < 0.0001), 35% (*p* < 0.001), 25% (*p* < 0.001), and 15% (*p* < 0.05) inhibition at 200 µg/mL, 100 µg/mL, 50 µg/mL, and 10 µg/mL, respectively (Figure 3D). The 10–30 DP FVF inhibited LPS induced TNF-α more effectively at lower than higher concentration, with percentage inhibition of 15% (*p* < 0.001), 15% (*p* < 0.001), and 10% (*p* < 0.05) at 10 µg/mL, 50 µg/mL, and 100 µg/mL concentrations, respectively (Figure 3F). However, DP FVF showed inhibition of 10% (*p* < 0.05) only at 10 µg/mL concentration (Figure 3E).

#### 2.3.3. *Macrocystis pyrifera* Fucoidan (MPF) and Its Derived Extracts Suppress LPS Stimulated TNF-α in Human THP-1 Cells

*Macrocystis pyrifera* (MPF) and 5–30 DP MPF exhibited profound inhibition of TNF-α in LPS induced THP-1 cells in a dose-dependent manner (Figure 3G,I). Both extracts significantly inhibited the TNF-α production in a similar manner. MPF showed percentage inhibition of 80% (*p* < 0.0001), 70% (*p* < 0.0001), 60% (*p* < 0.0001), and 30% (*p* < 0.001) at 200 µg/mL, 100 µg/mL, µg/mL, and 10 µg/mL respectively, (Figure 3G). The 5–30 DP MPF showed a similar but stronger pattern of inhibition of LPS induced TNF-α with 90% (*p* < 0.0001), 90% (*p* < 0.0001), 85% (*p* < 0.0001), and 70% (*p* < 0.0001) percentage inhibition at 200 µg/mL, 100 µg/mL, 50 µg/mL, and 10 µg/mL, respectively (Figure 3I). However, the 30+ MPF significantly inhibited the TNF-α more effectively at lower concentrations with 20% (*p* < 0.001), 15% (*p* < 0.001), and 10% (*p* < 0.001) percentage inhibition at 10 µg/mL, 50 µg/mL, and 100 µg/mL concentrations, respectively (Figure 3H). These results suggest that 5–30 DP MPF is the most potent inhibitor of LPS induced TNF-α in THP-1 monocytes along with MPF.

#### 2.3.4. Nutritional Grade Fucoidan from *Fucus vesiculosus* with High Polyphenol Content (FVC), *Ascophyllum nodosum* (ANF), and *Laminaria japonica* (LJF) Inhibit LPS Induced TNF-α in Human THP-1 Cells in a Dose-Dependent Manner

FVC is a nutritional fucoidan extract from *Fucus vesiculosus* with a high polyphenol content of 15.5%. FVC inhibited LPS induced TNF-α in THP-1 monocytes at higher concentrations in a dose dependant manner with a percentage inhibition of 35% (*p* < 0.001), 25% (*p* > 0.05), and 10% (*p* < 0.05) at the 200 µg/mL, 100 µg/mL, and 50 µg/mL concentrations, respectively, (Figure 3J). Similarly, *Ascophyllum nodosum* fucoidan (ANF) also exhibited significant suppression of TNF-α production in higher concentration with percentage inhibition of 25% (*p* < 0.05) and 10% (*p* < 0.05) at 200 µg/mL and 100 µg/mL, respectively (Figure 3K). However, fucoidan extract from *Laminaria japonica* (LJF) has showed the most significant inhibition of LPS induced TNF-α levels in a dose dependant manner in all the concentrations with 80% at 200 µg/mL (*p* < 0.0001), 70% at 100 µg/mL (*p* < 0.0001), 50% (*p* < 0.0001) at 50 µg/mL, and 20% (*p* < 0.001) at 10 µg/mL concentration (Figure 3L).

### 2.4. Effect of Fucoidan Extracts on LPS Induced Pro-Inflammatory Cytokine IL-1β in Human THP-1 Monocytes

#### 2.4.1. *Undaria pinnatifida* Fucoidan (UPF) and Its Derived Extracts Suppress LPS Induced IL-1β in Human THP-1 Cells

*Undaria pinnatifida* fucoidan (UPF) suppressed LPS induced IL-1β in a dose-dependent manner. The inhibition pattern is similar to the inhibition of LPS induced TNF-α in human THP-1 monocytes. The calculated percentage inhibition for (UPF) is 25% (*p* < 0.001), 20% (*p* < 0.001), 15% (*p* < 0.05), and 10% (*p* < 0.05) at concentrations of 200 µg/mL, 100 µg/mL, 50 µg/mL, and 10 µg/mL, respectively (Figure 4A). Additionally, DP UPF showed significant inhibition of 15% (*p* < 0.05) and 10% (*p* < 0.05) at concentrations of 10 µg/mL and 50 µg/mL, respectively (Figure 4B). However, 5–30 DP UPF only showed significant inhibition of 15% (*p* < 0.05) at 10 µg/mL concentration (Figure 4C). These results suggest that the whole extract from *Undaria pinnatifida,* UPF is the most effective extract against LPS induced IL-1β in human THP-1 monocytes as it showed strong inhibition across all the concentrations in a dose-dependent manner. However, DP UPF and fractionated DP UPF (5–30 DP UPF) were more potent at lower concentrations.

#### 2.4.2. *Fucus vesiculosus* Fucoidan (FVF) and Its Derived Extracts Suppress LPS Stimulated IL-1β in Human THP-1 Cells

Whole extract of *Fucus vesiculosus* (FVF) has been shown to suppress LPS induced IL-1β in human THP-1 monocytes in a dose-dependent manner. FVF has displayed the strongest inhibition of about 40% (*p* < 0.001) at a concentration of 200 µg/mL, 20% (*p* < 0.001) at 100 µg/mL, 15% (*p* < 0.05) at 50 µg/mL, and 10% (*p* < 0.05) at 10 µg/mL concentration (Figure 4D). In contrast to FVF, DP FVF has shown better inhibition in IL-1β production at lower concentrations, with 20% (*p* < 0.05) inhibition at 10 µg/mL of concentration, 15% (*p* < 0.05) at 50 µg/mL, and 10% (*p* < 0.05) inhibition at 100 µg/mL concentration (Figure 4E). Likewise, 10–30 DP FVF showed even stronger inhibition at lower concentrations 25% (*p* < 0.001), 20% (*p* < 0.05) and 15% (*p* < 0.05) at concentrations of 10 µg/mL, 50 µg/mL, and 100 µg/mL, respectively (Figure 4F). These results suggest that FVF can suppress LPS induced IL-1β in human THP-1 monocytes across all concentrations used in the study in a dose-dependent manner. In contrast, DP FVF and 10–30 DP FVF are more potent at lower concentrations, and their effectiveness is decreases with an increase in the concentration of fucoidan.

#### 2.4.3. *Macrocystis pyrifera* Fucoidan (MPF) and Its Derived Extracts Suppress LPS Induced IL-1β in Human THP-1 Cells

MPF and 30+ DP MPF inhibited LPS induced IL-1β in human THP-1 monocytes in a dose dependent manner in this study. MPF has shown 35% (*p* < 0.001), 30% (*p* < 0.001), and 25% (*p* < 0.05) inhibition at 200 µg/mL, 100 µg/mL and 50 µg/mL concentration, respectively (Figure 4G). Similar trend of inhibition is observed for 30+ DP MPF with 30% (*p* < 0.05), 20% (*p* < 0.05), 15% (*p* < 0.05), and 15% (*p* < 0.05) at 200 µg/mL, 100 µg/mL, 50 µg/mL, and 10 µg/mL concentrations, respectively (Figure 4H). In contrast, 5–30 MPF showed highest inhibition at lower concentrations; 30% (*p* < 0.05) inhibition at 10 µg/mL concentration, 25% (*p* < 0.001) at 50 µg/mL, and 20% (*p* < 0.05) at 100 µg/mL concentration were observed. These results suggest that 5–30 MPF is more potent at lower concentrations against LPS induced IL-1β production in human THP-1 monocytes.

#### 2.4.4. Fucoidan Extracts from *Fucus vesiculosus* with Higher Polyphenol Content (FVC), *Ascophyllum nodosum* (ANF), and *Laminaria japonica* (LJF) Inhibit LPS Induced IL-1β in Human THP-1 Cells in a Dose-Dependent Manner

FVC and ANF suppressed LPS induced IL-1β in human THP-1 monocytes at higher concentrations in a dose-dependent manner. FVC showed a percentage inhibition of 25% (*p* < 0.001), 15% (*p* < 0.001), and 10% (*p* < 0.05) at 200 µg/mL, 100 µg/mL, and 50 µg/mL concentration, respectively (Figure 4J); likewise, ANF showed a similar trend with 20% (*p* < 0.001) and 15% (*p* < 0.05) at 200 µg/mL and 100 µg/mL concentrations only (Figure 4K). However, LJF showed stronger inhibition of LPS induced IL-1β across all concentrations in a dose-dependent manner, with 40% (*p* < 0.0001), 35% (*p* < 0.001) 25% (*p* < 0.05), and 30% (*p* < 0.001) at 200 µg/mL, 100 µg/mL, 50 µg/mL, and 10 µg/mL concentrations (Figure 4L). These results suggest that all these three extracts (FVC, ANF, and LJF) exhibit strong properties to suppress LPS induced IL-1β in human THP-1 monocytes, with LJF exhibiting the potent inhibition among the three.

## 3. Discussion

Chronic inflammation is a root cause of many disease states, including heart disease, gastro-intestinal diseases, and cancer [28]. Dietary ‘non-drug’ interventions capable of reducing inflammation are highly desirable. Beyond the anti-inflammatory effects in a colitis model [14], fucoidan has shown promise in restoring gut mucosal function in high-performance athletes [29].

Fucoidan fractions of different sizes have differential effects, for example, in coagulation [30]. The methods used to extract fucoidan from the source algae can affect the molecular weight profile and composition of the fucoidan [31]. Although it is well established that fucoidan has numerous proven bioactivities, however, there are numerous factors that can effect these bioactivities, such as molecular weight (MW) of fucoidan [12], composition (e.g., monosaccharide composition, the degree of sulphation) [13], and structure (glycosidic linkages, the degree of branching and substitution, chain conformation, etc.) [14]. It is known that the fucoidan differs substantially between the source species, on each of these three parameters—the environment, the source seaweeds from where they were cultivated or harvested, and even the time of the year [15]. No two isolated fucoidans are exactly the same, even if they are extracted from the same seaweed species; they are all unique in their structure, composition, and bioactivities [16]. In this current research, we sought to examine how molecular size and depyrogenation of fucoidans from different species correlates with one aspect of inflammation—cytokine release in macrophages—to allow the optimization of fucoidan for dietary use. We evaluated the differential anti-inflammatory effects between nutritional grade extracts, derived depyrogenated (DP) extracts, and low molecular weight extracts (LMWF) in LPS-induced inflammatory response in PMBC and human THP-1 monocyte cell lines.

Cytotoxicity of the biomaterial is critical to evaluate prior to in vitro and in vivo tests of the material [32,33]. In line with previous studies, we found near negligible cell toxicity of fucoidan extracts (Figure 1) on the monocytes suggesting targeted cytokine inhibition without overt induction of immune suppression. Fucoidans have been known for their reduced cell toxicity in several cell-based studies. Several studies investigating the influence of fucoidans on the viability of cells, including THP-1, K562, HS-Sultan, NB4, BCBL-1, TY-1, HL-60, and U937, have reported either no or a reducing effect on the cell viability [34,35,36,37,38]. In a recent research, Peng Li et, 2017 in their investigation treated THP-1 cells with several concentrations of fucoidan. According to their findings, fucoidan did not exhibit significant cytotoxic effects on THP-1-derived macrophages up to 72 h of incubation, at a concentration of 200 µg/mL [39]. Similarly, Kaya Saskia et al., 2019 in their investigation showed a comprehensive analysis on the cell toxicity of several fucoidans on six different cell lines [40]. Their findings revealed that fucoidan from several species including *Fucus vesiculosus* showed no cytotoxic effects on these cell lines; moreover, they found that fucoidan increased the cell viability for some of the cell lines used in their study, and they finally concluded that fucoidan is non-toxic [40]. Cell viability data for all the fucoidan extracts is provided in (Appendix A).

TNF-α, IL-1β, and IL-6 have been well documented for their role in several inflammatory conditions, including lung inflammation, rheumatoid arthritis, and intestinal bowel diseases (IBD) [41,42,43,44,45,46,47,48]. During inflammatory conditions or in response to bacterial products such as LPS, macrophages secrete cytokines, including TNF-α and IL-1β [49]. These cytokines can promote the expression of vasodilators, nitric oxide and prostaglandins, to facilitate vasodilation and adhesion molecules such as E-selectin and intercellular adhesion molecule 1 (ICAM-1) to facilitate binding of leukocytes to the endothelium [50]. Furthermore, TNF-α induces NADPH oxidase oxidation, which enables the production of reactive oxygen species (ROS) [51]. Nitric oxide (NO) and reactive oxygen species (ROS) have a variety of inflammatory actions and are important regulators of immune responses [52]. Additionally, TNF-α and IL-1β induce an array of cytokines and chemokines in chronic inflammatory conditions and work synergistically to induce majority of inflammatory genes through a mechanism involving NF-κB pathway [53]. Among the cytokines regulated by NF-κB, the role of IL-6 is well recognised for its role in the chronic inflammatory responses. IL-6 carry out numerous of its functions via STAT3 activation [54,55,56]. Similarly, IL-6 can upregulate the expression of adhesion molecules. IL-6 also induces the differentiation of immune cells: CD4+ T cells into T helper 17 cells and CD8+ T cells into cytotoxic T-cells while inhibiting the differentiation of CD4+ T cells into regulatory T cells [55]. Notably, IL-6 is an important cytokine in a process termed leukocyte switch, where the major leukocytes during the early stages of inflammation are neutrophils which switch to monocytes during later stages of inflammation [55,57]. This switch is achieved by suppressing chemokines such as IL-8 and chemokine (C-X-C motif) ligand (CXCL)1 that would attract polymorphonuclear leukocytes and promoting chemokines such as monocyte chemotactic protein (MCP)-1 and MCP-2 that would then attract monocytes [58].

In a recent investigation from our research group, we utilized two fucoidan extracts including depyrogenated UPF and FVC in an *in vivo* mouse model [14]. We demonstrated that both extracts, when administered orally, could reduce the elevated levels of pro-inflammatory cytokines, including TNF-α, IL-1β, and IL-6, in the Dextran Sodium Sulfate (DSS) colitis mouse model and effectively ameliorated the experimental colitis [14]. In line with this, we now further demonstrate that fucoidan extracts from *Undaria pinnatifida, Fucus vesiculosus, Macrocystis pyrifera, Ascophyllum nodosum,* and *Laminaria japonica* with different molecular weight profiles (Table 2) have significant anti-inflammatory effects in an *in vitro* cell model of LPS-induced PBMC cells and THP-1 cells.

Initial studies were carried out on PBMCs using only two nutritional grade extracts of fucoidan from *Undaria pinnatifida* and *Fucus vesiculosus*. These are the two trademark compounds from Marinova Pty Ltd. (Cambridge, TAS, Australia), that were initially tested for their anti-inflammatory activity in PBMC cells derived from three individuals. Both extracts showed significant attenuation of pro-inflammatory cytokines including TNF-α and IL-6, (Figure 2A,C–F) in PBMC cells in a dose dependent manner. Based on these results, further experiments were conducted to produce depyrogenated and lower molecular weight fractions for cataloguing of these extracts. Since most of the compounds were untested previously, we used an *in vitro* cell based pro-inflammatory screening model using THP-1 cells to identify the most potent compounds among these fractions.

All fucoidan extracts showed significant attenuation of LPS induced pro-inflammatory cytokine production, including TNF-α and IL-1β in human THP-1 cells, with some extracts almost completely reversing the LPS effect. There was a marked and consistent size-based effect of these extracts on the LPS induced immune response in THP-1 cells, with nutritional grade extracts of fucoidan demonstrated effects mainly at higher concentrations in a dose dependent manner while depyrogenated extracts and LMWF extracts showing significantly higher inhibition mostly at lower concentrations for most of the extracts. This effect was consistent across fucoidan extracted from different algae. The anti-inflammatory effects of fucoidan have been reported in several studies, for example, Jeong et al., reported decreased secretion of PGE2 in RAW 264.7 cells pre-treated with LPS; moreover, fucoidan hindered the nuclear accumulation of NF-κB subunit. Fucoidan from *Fucus vesiculosus* also diminished the secretion of TNF-α and IL-1β [59]. Similarly, Ni et al. reported decreased NO production, iNOS and COX2 expression and impeded the NF-κB signaling in LPS treated RAW 264.7 upon fucoidan treatment [60]. In line with these studies, our results in (Figure 3A,D,G,J–L) and (Figure 4A,D,G,J–L), nutritional grade extracts from *Undaria pinnatifida, Fucus vesiculosus, Fucus polyphenol, Macrocystis pyrifera, Ascophyllum nodosum,* and *Laminaria japonica* have shown consistent down regulation of these cytokines in THP-1 cells in a dose-dependent manner. During an inflammatory response, activated TNF-α and IL-1β work synergistically to activate both NF-κB and STAT-3 signaling pathways, ultimately leading to the activation of downstream inflammatory cascade. Our results suggest that the attenuation of both TNF-α and IL-1β by nutritional grade fucoidan extracts is possibly via downregulation of the NF-κB pathway, which is consistent with the previous studies [59,60,61].

Fucoidan from *Macrocystis pyrifera,* has previously been utilized by Wei Zhang et al., 2015 to evaluate its effects on human neutrophil apoptosis and on mouse natural killer (NK) cell, (dendritic cells) DC, and T cell activation *in vivo* [62]. According to their findings *Macrocystis pyrifera,* has shown to be strong immune stimulator. It significantly enhanced the NK cells production, promoted the maturation of DC cells and induced the production of pro-inflammatory cytokines including TNF-α, and IL-6. Likewise, Wei Zhang et al., 2021 reported that *Macrocystis pyrifera,* strongly upregulated the expression of CD 80, CD 83, CD 86, MHC class I and MHC class II, induced the expression of pro-inflammatory cytokines expression (TNF-α, and IL-6) and promoted the human monocyte-derived dendritic cells (MODC) and human peripheral blood DC (PBDC) activation [63]. In contrast to their findings, here we provide evidence for the first time on fucoidan extracted from *Macrocystis pyrifera*, which we found to have strong anti-inflammatory effects in our LPS induced *in vitro* model. All three extracts of *Macrocystis pyrifera,* were effective against LPS induced pro-inflammatory cytokines. The lowest molecular weight extract of *Macrocystis pyrifera,* was the most potent inhibitor at lower concentrations of both TNF-α and IL-1β in LPS stimulated human THP-1 cells. This suppressive action suggests a potential therapeutic role.

In the current investigation, for the first time, four novel depyrogenated extracts have been studied from *Undaria pinnatifida* and *Fucus vesiculosus* (Figure 3B,C,E,F and Figure 4B,C,E,F). Even though fucoidans have the potential for use in diverse and clinically important applications. A considerable challenge in exploiting the benefits of these molecules is the presence of pyrogenic agents (endotoxins) in the extracts. primarily, where the disease indication requires parenteral administration. The direct application of these extracts to a wound or a surgical site, or implantation within the body, is contraindicated due to the linked dangers of pyrogen-induced fever, toxic shock, and even death. The process of removal of these pyrogens is called depyrogenation. Although in our results the depyrogenated extracts have shown reduced suppression of inflammatory cytokine, an interesting reverse dose dependent trend in the suppression of TNF-α production is observed. This reduced suppression at higher concentration and greater suppression at lower concentration could be associated to extra pure fucoidan extracts obtained after depyrogenation method, as a result these fucoidan extracts interact with cell surface receptors more effectively at lower concentrations. We understand that the use of depyrogenated fucoidan extracts for their anti-inflammatory has been reported for the first time and such reverse dose dependent effect can further be elaborated using future experiments with much lower concentration, thus generating a better dose dependent response for these extracts. From this study, we can conclude that depyrogenated fractions have better anti-inflammatory properties than the original fractions, although these outcomes must be checked in the pre-clinical disease models.

Similarly, LMWF depyrogenated extracts show the similar trend, but with most of the LMWF extracts showing even increased suppression at lower concentrations. It could be related to the lower molecular weight of these extracts that is directly related to better absorption of these extracts through the cell surface. Similarly, previously stated that LMWF are efficient molecules due to their better absorption and rapid mode of action. However, there long-term effects in the cell environment have been debatable in some recent investigations. Nevertheless, LMWF have been off immense interest in the development of the pharmaceutical and biotech industries due to their promising properties. Further experiments with a focus on generating a dose response curve should be performed and anti-inflammatory effect of these low molecule should be studied at further lower concentrations.

Here, our work indicates that fucoidan extracts reduced the levels of pro-inflammatory markers *in vitro* both in PBMC cells and in THP-1 cells. The molecular mechanism through which fucoidan attenuates the pro-inflammatory markers remains unclear. There are two potential mechanisms through which fucoidan mitigates these inflammatory responses. One possible mechanism is via blocking of the macrophage’s maturation directly, or secondly, through downregulation of cytokines released by the macrophages and other innate cells during active inflammatory conditions ultimately attenuates the NF-κB signaling pathway. However, macrophages appear to be the leading players that initiate inflammatory processes and thus are the primary targets in inflammatory-related diseases [22].

THP-1 cells are derived from the peripheral blood from a patient with acute monocytic leukemia [64,65], whereas peripheral blood mononuclear cells (PBMCs) are blood cells with round nuclei, such as monocytes, lymphocytes, and macrophages, isolated from the peripheral blood of healthy donors [25,64]. Both THP-1 and PBMC cells were chosen in this study as they have been commonly used in research investigating immune function in monocytes and macrophages, including the secretion of pro-inflammatory cytokines. The THP-1 cell line has certain advantages over PBMCs, such as easier acquisition, faster doubling time, and being stored long-term in liquid nitrogen to recover later without impacting cell viability. However, the malignant background of the THP-1 could produce an immune response that does not exactly translate into a similar response *in vivo* conditions. Anita et al. found that THP-1 monocytes produced lower pro-inflammatory cytokines upon stimulation with LPS than fresh PBMC monocytes. LPS treatment of THP-1 and PBMCs showed no signs of phenotypic change in THP-1 cell morphology in 24 h incubation. John D. Widdrington et al. reported in their finding that LPS did not adversely affected THP-1 cell viability; additionally, their study revealed no morphological change, adherence capacity, and confirmed the absence of differentiation into macrophage-like cells compared with the positive control treated with PMA for 72 h [65]. Similarly, Thaize Quiroga Chometon et al. reported that GM-CSF and IL-4 combination resulted in PBMC derived monocytes differentiation and maturation [66], which is in line with our work.

This study, so far to our knowledge, is one of the first comprehensive studies investigating such a vast number of clearly defined and well-characterized fucoidan extracts. The key observation that lower molecular weight extracts appear to have high specificity at low concentrations has implications for maximizing dietary fucoidan’s effects in inflammatory conditions. The ionic nature of fucoidan plays an important role in its application. The negative charge of the molecule results from the presence of sulfate residues in the C-2 and C-4 positions, intermittently in C-3, allowing the complex formation with other oppositely charged molecules [67]. Fucoidan extracts may aggregate or interact with other components in the medium during incubation, preventing their efficient interaction with monocytes. LMWF, due to their smaller size may accelerate drug absorption and possess better access to cell surface receptors. However, more pre-clinical studies are needed to understand how these extracts work as anti-inflammatory agents in animal models, which will pave the way for human clinical studies.

## 4. Materials and Methods

All fucoidan extracts used in this research were extracted as per commercial targeted filtration techniques. Furthermore, fucoidan purity estimations require multiple inputs which were determined by means of spectrophotometric assays. A hydrolyzed sample was assessed by the phenol–sulfuric technique from Dubois for total carbohydrate [68,69], while the uronic acid concentration was measured in the presence of 3-phenylphenol by a spectrophotometric examination of the hydrolyzed product, based on Filisetti-Cozzi and Carpita technique [69]. The amount of sulfate was determined spectrophotometrically using a BaSO_4_ precipitation technique based on Dodgson’s work [70]. The molecular weight profile was calculated using gel permeation chromatography and a size-exclusion column, and the results were compared to Dextran standards. All the fucoidan extracts were obtained from Marinova (Cambridge, TAS, Australia) and the chemical composition of each extract is described in Table 2.

### 4.1. Cell Viability Assay

The MTT viability assay was performed to measure the cell viability, as previously described [14]. Briefly, MTT (3-[4,5-dimethylthiazol-2-yl]-2,5,-diphenyltetrazolium bromide) (Sigma Aldrich, Castle Hill, NSW, Australia) was dissolved phosphate-buffered saline (PBS, pH 7.4) at a final concentration of 5 mg/mL. THP-1 cells (1 × 10^5^ cells/well) were incubated with four concentrations of fucoidan extracts (200 µg/mL, 100 µg/mL, 50 µg/mL, and 10 µg/mL) in a U bottom 96-well tissue culture plate. The final volume was 100 μL, including both the cell suspension and the fucoidan extracts. Following an incubation period of 48 h at 37 °C, 5% CO_2_, MTT reagent (20 μL/well) was added, and the plate was incubated for an additional 3 h in the incubator. The cells were then washed three times in PBS. Finally, MTT formazan crystals were dissolved in 100 μL of dimethyl sulfoxide (DMSO), and the absorbance was measured on a microplate reader (Tecan pro 2000 plate reader) at 570 nm. The percentage of the viable cells was calculated using the following formula: (%) = [100 × (sample abs)/(control abs)].

### 4.2. PBMC Cell Culture, Fucoidan Treatment, and Stimulation

The study (Project PBMC-JHF-1) was conducted according to the guidelines of the Declaration of Helsinki, and approved by the Institutional Review Board (or Ethics Committee) of Washington Biotechnology IACUC (Approval number 18–342 on August 6, 2018). For human blood mononuclear cells (PBMC) studies, healthy donors (*n* = 3) with no clinical history were selected to donate blood under authorized supervision from subjects who had given their consent. PBMCs are blood cells with a heterogeneous cell population, comprising of monocytes, lymphocytes, and macrophages [71,72]. PBMCs were separated in a single-step density centrifugation technique, as described previously [14]. Briefly, peripheral blood diluted in Hanks Balanced salt solution (HBSS) was layered over Ficoll and centrifuged for 30 min at 400× *g* (20 °C). Cells harvested from the interface and were washed 3 times in HBSS and re-suspended in a culture medium consisting of RPMI 1640 supplemented with 10% (*v*/*v*) heat-inactivated FBS and penicillin-streptomycin (100 IU/mL and 1 µg/mL) in a 96-well plate and were pre-treated with three concentrations of UPF and FVF (10 µg/mL, 50 µg/mL, 100 µg/mL) for 4 h. Followed by the pre-treatment, PBMC cells were stimulated with Lipopolysaccharide (LPS) (*Escherichia coli*, Sigma) for 24 h with a final concentration of 200 ng/well) to induce the secretion of TNF-α, IL-1β, and IL-6. Both extracts of fucoidan are water soluble, and the stock solution was dissolved in a cell culture medium to be added to the cells. Untreated PBMC cells in the RPMI medium without treatment and PBMC cells stimulated with LPS alone served as controls. Following LPS incubation for 24 h, cells were centrifuged at 300× *g* for 10 min, supernatant was collected and froze at −80 °C until further used for ELISA.

### 4.3. THP-1 Cell Culture, Fucoidan Treatment, and LPS Stimulation

The THP-1 human monocytic cell line, derived from acute monocytic leukaemia were obtained from Sigma Aldrich (Castle Hill, NSW, Australia), was maintained in RPMI 1640 medium (Sigma) supplemented with 10% (*v*/*v*) heat-inactivated FBS and penicillin-streptomycin (100 IU/mL and 1 µg/mL) and incubated at, 5% CO_2_. Cells were seeded at a density of 1 × 10^5^ per well in a round bottom 96 well plate purchased from Thermo Fisher Scientific Inc, (Scoresby, VIC, Australia), and were pre-treated with the desired concentrations of fucoidan extracts (200 µg/mL, 100 µg/mL, 50 µg/mL, 10 µg/mL) and incubated at 37 °C, 5% CO_2_ for 24 h. Pre-treated cells with fucoidan extracts were stimulated with 1 µg/mL of LPS from *Escherichia coli* (L4391, Sigma) for 24 h. Monocytes were stimulated with LPS without fucoidan treatment serving as positive controls, while monocytes without LPS and no treatments served as negative controls.

### 4.4. Enzyme-Linked Immunosorbent Assay ELISA

Levels of TNF-α, IL-1β, and IL-6 were measured from the supernatant by ELISA using capture and detection antibodies from Peprotech (Sydney, NSW, Australia) and BD Biosciences (Sydney, NSW, Australia) per the manufacturer’s instructions. The amount of protein secreted was normalised to the amount secreted by macrophages treated with LPS alone. Each experiment was performed at least 3 times. Plate reading and curve fittings were performed on a plate reader (TECAN infinite M200, Männedorf, Switzerland) using iControl software (TECAN Group Ltd.).

### 4.5. Statistical Analysis

Data represented in this study are as means ± SEM. Statistical analyses were carried out using parametric one-way analysis of variance (ANOVA) followed by Tukey’s multiple comparison test. A *p*-value of <0.05 was considered significant, with * *p* < 0.05, ** *p* < 0.001, *** *p* < 0.0001, and **** *p* < 0.00001. Percentage inhibition was calculated for each significant value using the formula 100 − (drug treated cells/LPS control) * 100.

## 5. Conclusions

Fucoidan extracts are known to exert anti-inflammatory effects. We have successfully catalogued many fucoidan extracts with respect to their efficacy in macrophages and PBMCs in downregulating major pro-inflammatory cytokines, potential targets in human inflammation. In this study, *Macrocystis pyrifera* extracts were shown for the first time to have significant anti-inflammatory effects. The low molecular weight fucoidan LMWF extracts within *Macrocystis pyrifera* have shown strong anti-inflammatory effects at lower concentrations. Further research would provide more insights about the mechanism of action with a view to testing for therapeutic applications as an anti-inflammatory medication.

## Figures and Tables

**Figure 1 marinedrugs-19-00702-f001:**
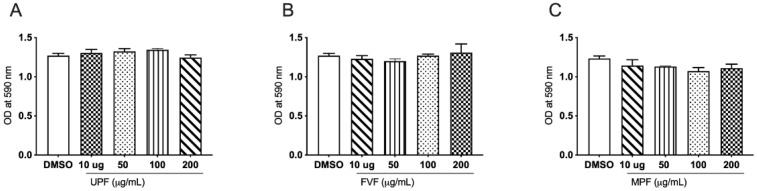
Effects of fucoidan extracts on THP-1 cell viability. Cells were treated with the specified concentrations of fucoidan extract UPF (**A**), FVF (**B**) and MPF (**C**) for 48 h and assessed by MTT reduction assays. Results expressed OD of treated cells vs. vehicle control. Data represented as means ± SEM. Statistical analyses were carried out using one-way ANOVA followed by Tukey’s multiple comparison test of three independent experiments run in triplicates (*n* = 3). A *p*-value of <0.05 was considered significant.

**Figure 2 marinedrugs-19-00702-f002:**
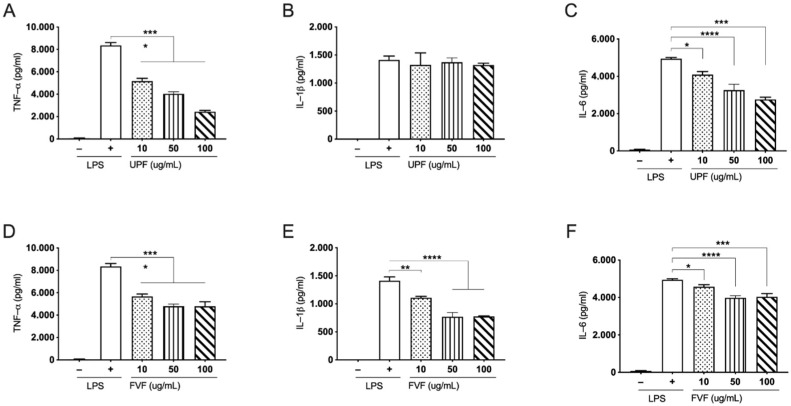
Pre-treatment of fucoidan extracts on pro-inflammatory cytokines, TNF-α, IL-1β, and IL-6 in LPS-stimulated human PBMC cells. Cells were pre-treated with the indicated doses UPF (**A**–**C**) and FVF (**D**–**F**) for 1 h followed by LPS stimulation for 24 h. The amounts of TNF-α, IL-1β, and IL-6 were measured from the supernatant using ELISA. Data represented as means ± SEM. Statistical analyses were carried out using one-way ANOVA followed by Tukey’s multiple comparison test of three independent experiments run in triplicates (*n* = 3) A *p*-value of <0.05 was considered significant. * *p* < 0.05, ** *p* < 0.001, *** *p* < 0.0001 and **** *p* < 0.00001.

**Figure 3 marinedrugs-19-00702-f003:**
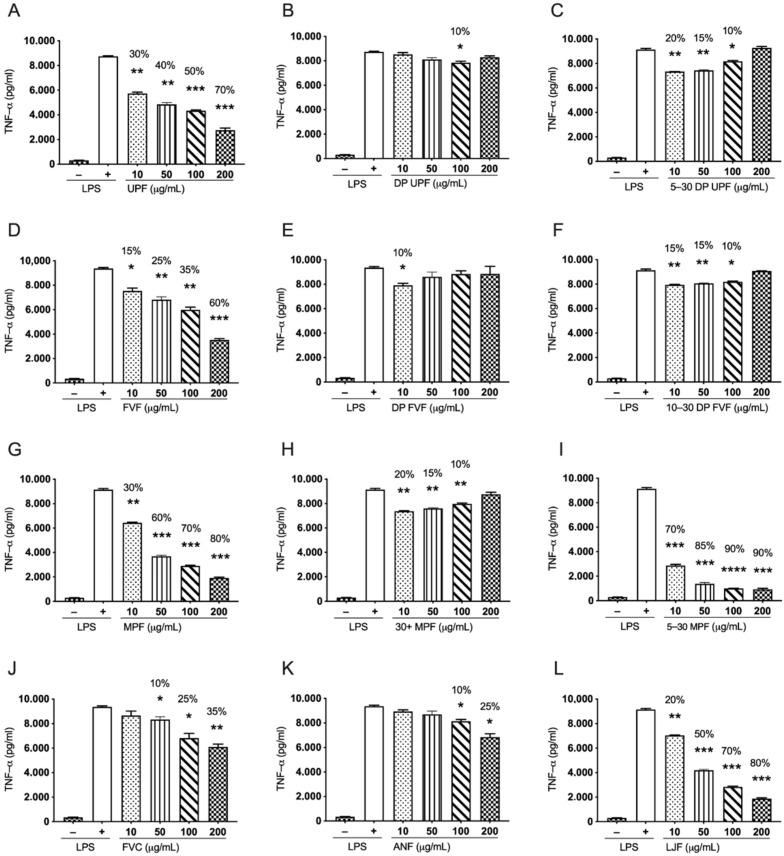
Effects of multiple fucoidan extracts on LPS induced pro-inflammatory cytokine TNF-α in human THP-1. Cells were pre-treated with several fucoidan extracts. UPF and its derived extracts (**A**–**C**), FVF and its derived extracts (**D**–**F**), MPF and derived extracts (**G**–**I**), FVC (**J**), ANF (**K**) and LJF (**L**) with indicated doses (10, 50, 100, and 200 µg/mL) for 24 h before LPS treatment (0.5 µg/mL), supernatants were isolated, and the amounts of TNF-α was measured. Data represented as means ± SEM. Statistical analyses were carried out using one-way ANOVA followed by Tukey’s multiple comparison test of three independent experiments run in triplicates (*n* = 3). A *p*-value of <0.05 was considered significant. * *p* < 0.05, ** *p* < 0.001, *** *p* < 0.0001, and **** *p* < 0.00001.

**Figure 4 marinedrugs-19-00702-f004:**
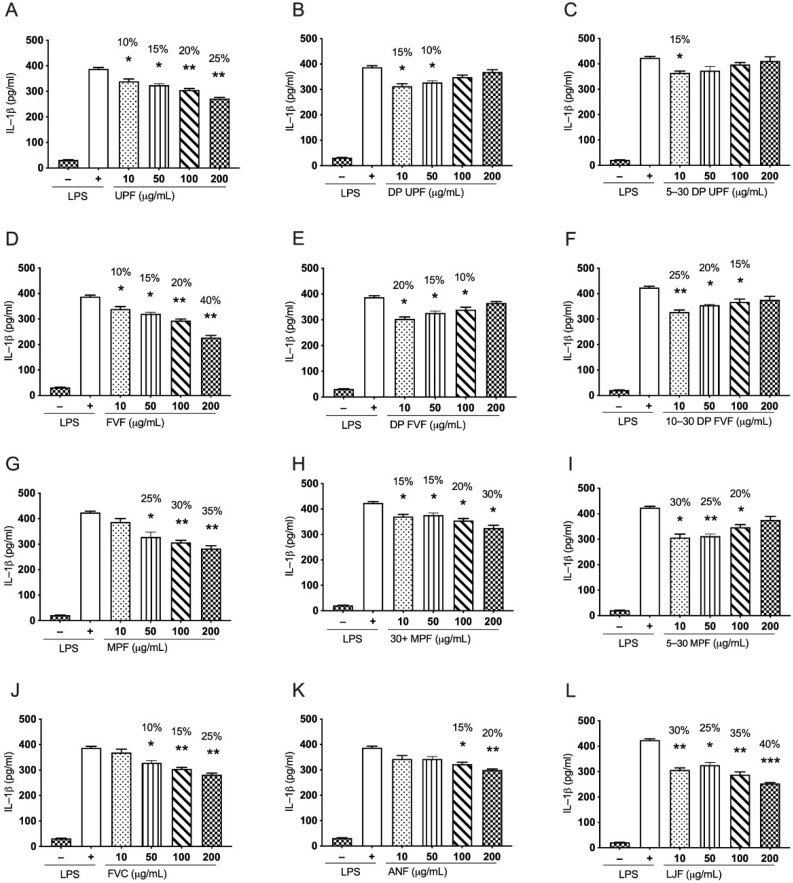
Effects of multiple fucoidan extracts on LPS induced pro-inflammatory cytokine, IL-1β in human THP-1 monocyte cells. Cells were pre-treated with several fucoidan extracts. UPF and its derived extracts (**A**–**C**), FVF and its derived extracts (**D**–**F**), MPF and derived extracts (**G**–**I**), FVC (**J**), ANF (**K**) and LJF (**L**) with indicated doses (10, 50, 100, and 200 µg/mL) for 24 h before LPS treatment for 24 h (0.5 µg/mL), supernatants were isolated, and the amounts of TNF-α were subsequently measured. Data represented as means ± SEM. Statistical analyses were carried out using one-way ANOVA followed by Tukey’s multiple comparison test of three independent experiments run in triplicates (*n* = 3). A *p*-value of < 0.05 was considered significant. * *p* < 0.05, ** *p* < 0.001, and *** *p* < 0.0001.

**Table 1 marinedrugs-19-00702-t001:** List of Fucoidan extracts with purity described.

Code	Species	Origin	Description
UPF	*Undaria pinnatifida*	South American	≥85% purity
DP UPF	*Undaria pinnatifida*	South American	≥90% purity and depyrogenated
5–30 DP UPF	*Undaria pinnatifida*	South American	≥90% purity, depyrogenated and 5–30 kDa
FVF	*Fucus vesiculosus*	Canadian	≥90% purity
DP FVF	*Fucus vesiculosus*	Canadian	≥90% purity and depyrogenated
10–30 DP FVF	*Fucus vesiculosus*	Canadian	>90% purity, depyrogenated and 10–30 kDa
FVC	*Fucus vesiculosus*	Canadian	≥85% purity
MPF	*Macrocystis pyrifera*	South American	≥85% purity
30+ DP MPF	*Macrocystis pyrifera*	South American	≥85% purity, ≥30 kDa and depyrogenated
5–30 DP MPF	*Macrocystis pyrifera*	South American	≥85% purity, 5–30 kDa and depyrogenated
ANF	*Ascophyllum nodosum*	Canadian	≥85% purity
LJF	*Laminaria japonica*	Japan	≥85% purity

**Table 2 marinedrugs-19-00702-t002:** Chemical composition, molecular weight distribution, and carbohydrate profile of fucoidan extracts used in the current study. All extracts are polydisperse with molecular weight ranges from 5 kDa to 600 kDa. All nutritional grade extracts having molecular weight greater than 30 kDa considered nutritional grade. Extracts having molecular weight below 30 kDa were considered LMWF.

Co	Total Carbohydrates (%)	Uronic Acid (%)	Polyphenols (%)	SO_4_ (%)	Cations (%)	Peak MW (kDa)
UPF	54.2	4.1	<2.5	24.9	8.0	40.4
DP UPF	64.0	0.9	<2.5	31.0	6.8	134.2
5–30 DP UPF	61.3	0.7	<2.5	27.6	6.0	8.2
FVF	66.5	3.6	2.5	25.9	6.2	35.8
DP FVF	67.5	2.0	<2.5	26.0	11.2	54.0
10–30 DP FVF	71.5	2.2	<2.5	26.9	8.8	21.3
FVC	59.6	6.2	15.5	14.0	5.7	48.8
MPF	51.1	6.1	<2.5	25.7	7.9	66.0
30+ DP MPF	56.9	5.7	<2.5	23.0	10.8	32.5
5–30 DP MPF	54.6	7.1	<2.5	19.7	10.8	17.4
ANF	66.6	4.1	17.5	24.2	5.0	115.5
LJF	55.2	23.8	<2.5	18.6	5.0	597.1

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
