# Peer review of "Anti-Inflammatory Activity of Fucoidan Extracts In Vitro"

_marinedrugs, 2021, doi:10.3390/md19120702_

Round 1

Reviewer 1 Report

In this study, fucoidans from several algal species have been assessed for their ability to modulate LPS-induced production of inflammatory cytokines from human PBMCs and THP-1 cells.

The central section of the paper, centred around figures 3 and 4, concerns suppression of TNF-α and IL-1β production by the fucoidan samples from Undaria, Fucus and Macrocystis species. The fucoidan preparations as received all had dose-dependent ability to suppress TNF-α production, but depyrogenated samples had much reduced suppression and showed a ‘reverse dose dependence’ with higher concentrations being less effective. This remarkable reduction of anti-inflammatory activity by pyrogen removal is hardly discussed in the paper, and the details of the depyrogenation procedure are not presented. Low molecular weight fractions of the fucoidans tell the same story; the Macrocystis fraction, not depyrogenated, has strong activity whereas the LMW fractions of depyrogenated Fucus and Undaria fucoidans have less activity and a reversed dose response relationship. It is not at all easy to interpret these observations, but they should not be ignored but fully discussed and explained.

Given the controversy over the pro- and anti-inflammatory activities of fucoidans described in lines 279-296, it is surprising that the authors ignore this aspect of their results. Perhaps rather than use depyrogenated samples, the authors should consider testing their fucoidans for LPS-like activity directly, simply using the same experimental setup but without LPS. This is in any case an important control experiment that should be performed.

A minor comment: results in the diagrams are presented as means with error bars based on SEM. Please include also the value of n, the number of independent measurements, for these results.

Author Response

Reviewer 1

Comments and Suggestions for Authors

In this study, fucoidans from several algal species have been assessed for their ability to modulate LPS-induced production of inflammatory cytokines from human PBMCs and THP-1 cells.

The central section of the paper, centred around figures 3 and 4, concerns suppression of TNF-α and IL-1β production by the fucoidan samples from Undaria, Fucus and Macrocystis species. The fucoidan preparations as received all had dose-dependent ability to suppress TNF-α production, but depyrogenated samples had much reduced suppression and showed a ‘reverse dose dependence’ with higher concentrations being less effective. This remarkable reduction of anti-inflammatory activity by pyrogen removal is hardly discussed in the paper, Low molecular weight fractions of the fucoidans tell the same story; the Macrocystis fraction, not depyrogenated, has strong activity whereas the LMW fractions of depyrogenated Fucus and Undaria fucoidans have less activity and a reversed dose response relationship. It is not at all easy to interpret these observations, but they should not be ignored but fully discussed and explained

Answer

Thank you for your observation.  As per the advice from the reviewer, we have now incorporated more discussion around the theme We have updated the information in the text, from line 400-431.

and the details of the depyrogenation procedure are not presented.

Answer

Depyrogenation procedure was carried out as per the patented protocol by Marinova ((Patent # 10,149,874)

Given the controversy over the pro- and anti-inflammatory activities of fucoidans described in lines 279-296, it is surprising that the authors ignore this aspect of their results. Perhaps rather than use depyrogenated samples, the authors should consider testing their fucoidans for LPS-like activity directly, simply using the same experimental setup but without LPS. This is in any case an important control experiment that should be performed.

Answer

In this screening study, our main plan was to look at the effect of these fucoidan and the derived extracts in an active inflammatory condition, as observed in several inflammatory diseases. Although not discussed, our larger aim was to use these extracts in inflammatory bowel disease pre- clinical model, to ascertain the effect on infections from opportunistic pathogens such as E. coli that expresses LPS on its cell surface. We do agree with the referee the importance of looking at the effects of fucoidan on its own, though in these experiments we planned to categorise these molecules based on their anti-inflammatory properties. Additionally, we have incorporated information in the text in line 281-290 to address this comment by the reviewer.

A minor comment: results in the diagrams are presented as means with error bars based on SEM. Please include also the value of n, the number of independent measurements, for these results.

Answer

Thank you for this observation, we have now updated all figure legends with required information, including value of n, and the number of independent measurements.

Reviewer 2 Report

In their article, the authors have investigated the effect of several fucoidans on the pro-inflammatory activation of a macrophage cell line and on PBMCs. While the study is of interest, a variety of concerns need to be addressed before publication can be considered.

Methods

The authors have used THP-1 cells and PBMCs. It is important to notice that PBMCs are not only macrophages. This needs to be addressed and discussed. Furthermore, it is not clear whether the different experiments with PBMCs were conducted with PBMCs of the same donor (technical replicates) or of different donors (biological replicates). This needs to be described in the method section. The sample size needs to be given (in the figure legends).

Also, the authors claim that they have tested the many fucoidans in THP-1 cells and PBMCs (e.g. line 266), while in fact, in PBMCs, only two different fucoidans were tested. This needs to be corrected in the text. Why were these particular fucoidans tested in PBMCs? They were not the most effective in THP-1.

The fucoidans need to be better described. When stating their origin, in addition to species, the geographical origin should be given. The extraction methods should be mentioned – as employees of Marinova are involved in this article that information should be available. Despite claimed otherwise, the molecular weight is of the fucoidans is not given and it is not defined what is actually considered “high molecular weight” or “low molecular weight. Molecular weight and/or the range of molecular weight of the specific fucoidans should be provided.

The authors tested toxicity of fuocidan only for THP-1, what about PBMCs?

The authors tested the toxicity of fucoidan, but they also should check on the toxicity of LPS.

Further comments

Introduction

Line 43, “these distinctive properties”, if the authors are referring to the variability of the biological activities of fucoidan as pointed out in the sentence before, these are actually quite an obstacle for drug development.

Discussion

The discussion needs to be revised to provide references to published studies and, indeed, discuss the findings.

Cell toxicity, please discuss the lack of toxicity for this cell line with regard to what has been published on fucoidan toxicity in the literature.

Line 276, following, it would be interesting to learn why fucoidan could have these different effects? Could it be due to the molecular properties of the fucoidans? Or the different model systems? It is well known that fuocidans have different effects in different systems (e.g. the same fucoidan being pro- and antiangiogenic in different cell systems). Please discuss what could be responsible for these differences.

Line 328, following, do THP-1 mature under LPS treatment? Please explain/elaborate. What about PBMCs?

Author Response

Please see attached file displaying all the changes

Round 2

Reviewer 1 Report

No further comments, much improved in the new version.

Reviewer 2 Report

The authors have addressed all concerns to the reviewer's satisfaction.